# EatWellNow: Formative Development of a Place-Based Behavioral “Nudge” Technology Intervention to Promote Healthier Food Purchases among Army Soldiers

**DOI:** 10.3390/nu14071458

**Published:** 2022-03-31

**Authors:** Jared T. McGuirt, Alison Gustafson, Alice S. Ammerman, Mary Tucker-McLaughlin, Basheerah Enahora, Courtney Moore, Danielle Dunnagan, Hannah Prentice-Dunn, Sheryl Bedno

**Affiliations:** 1Department of Nutrition, School of Health and Human Sciences, University of North Carolina Greensboro, Greensboro, NC 27412, USA; brenahor@uncg.edu; 2Department of Dietetics and Human Nutrition, University of Kentucky, Lexington, KY 40506, USA; alison.gustafson@uky.edu; 3Department of Nutrition, University of North Carolina at Chapel Hill, Chapel Hill, NC 27599, USA; alice_ammerman@unc.edu; 4School of Communication, East Carolina University, Greenville, NC 27858, USA; tuckermclaughlinm@ecu.edu; 5Womack Army Medical Center, Fort Bragg, NC 28310, USA; courtney.a.moore35.mil@mail.mil (C.M.); danielle.l.dunnagan.mil@mail.mil (D.D.); sbedno@gmail.com (S.B.); 6UNC Lineberger Cancer Center, University of North Carolina at Chapel Hill, Chapel Hill, NC 27599, USA; hannah_prentice-dunn@med.unc.edu

**Keywords:** food environment, military, digital technologies, geofencing, beacons

## Abstract

Approximately 17% of military service members are obese. Research involving army soldiers suggests a lack of awareness of healthy foods on post. Innovative approaches are needed to change interactions with the military food environment. Two complementary technological methods to raise awareness are geofencing (deliver banner ads with website links) and Bluetooth beacons (real-time geotargeted messages to mobile phones that enter a designated space). There is little published literature regarding the feasibility of this approach to promote healthy behaviors in retail food environments. Thus, we conducted a formative feasibility study of a military post to understand the development, interest in, and implementation of EatWellNow, a multi-layered interactive food environment approach using contextual messaging to improve food purchasing decisions within the military food environment. We measured success based on outcomes of a formative evaluation, including process, resources, management, and scientific assessment. We also report data on interest in the approach from a Fort Bragg community health assessment survey (*n* = 3281). Most respondents agreed that they were interested in receiving push notifications on their phone about healthy options on post (64.5%) and that receiving these messages would help them eat healthier (68.3%). EatWellNow was successfully developed through cross-sector collaboration and was well received in this military environment, suggesting feasibility in this setting. Future work should examine the impact of EatWellNow on military service food purchases and dietary behaviors.

## 1. Introduction

A 2020 report suggests that one in five military service members are classified as obese (BMI of 30.0 and above) and have difficulty meeting the Army Body Composition Program (ABCP) standard [1]. A 2016 study found that active-duty military service members with obesity were 33% more likely to suffer musculoskeletal injury, leading to over 3.6 million injuries between 2008 and 2017 [2]. This problem also causes a substantial economic burden for the Department of Defense, which spends over $1 billion a year on healthcare costs among active-duty service members, veterans, and their families [3]. Additionally, it is estimated that there are over 650,000 days of lost work per year among active-duty military due to obesity-related health issues [4].

Research suggests that soldiers have low adherence to the Dietary Guidelines for Americans [5]. Obesity and poor dietary choices have been associated with poor attentiveness, reduced vision, adversarial work relationships, and reduced physical fitness [6,7,8]. A high level of body fat has been shown to harm performance in several military critical tasks [9,10]. Thus, strategies are needed to optimize diet-related health in military service members.

US military installations often have abundant unhealthy food options, including fast food, which has led to military officials seeking changes to improve the food environment [11,12,13,14,15]. Through programs like the DoD Go For Green initiative [16], Army’s Performance Triad program [17], Healthy Base Initiative [13], and Holistic Health and Fitness [18], the military has made efforts to improve the food environment in military installations. While improving availability is essential, service members must be aware of this availability and feel accommodated during their food purchase (i.e., the environment is set up to make shopping easy and enjoyable) [19]. Previous research has found that US Army Soldiers perceive a lack of healthy options on the installation, despite these efforts [20].

Thus, there is a need for approaches that make military service members more aware of healthy options and “nudge” them to purchase and consume these healthier options. Nudging is a key component of Behavioral Economics as a part of choice architecture [21]. Behavioral Economics uses choice architecture to design environments to influence consumer decision-making [21]. A practical approach uses cues or environmental triggers that remind customers to make healthier choices [21]. A recent meta-analysis of choice architecture nudging interventions found that choice architecture interventions promote behavior change, particularly when influencing food choices [22].

One emerging approach is “geofencing”, a real-time targeted marketing approach [23,24,25]. A geofence is a virtual perimeter in a real-world geographic area. These virtual perimeters can be established in multiple ways, including using cellular data signals [25,26]. A person crosses into a virtual geofence with their mobile device (i.e., smartphone). If the location service on their device is on (in an app or website that they are on), the person’s location can be detected to receive the geofence messaging. From this point, as part of the ”user audience”, this person can receive a banner, display, and/or video advertisement. A similar but different approach to geofencing is geotargeted messaging through Bluetooth beacons, which are hardware transmitters that broadcast their identifier to nearby portable electronic devices (e.g., smartphones) when within a set distance (i.e., 10 feet) [27]. Once within that distance, it can trigger an associated mobile phone app to post a push notification message to the smartphone.

These mobile phone messaging approaches are rapidly increasing in the retail sector as a targeted advertisement technique for customers within a particular geographic area [26]. Research has found that 60% of American adult consumers look for local information on their mobile devices, 40% look for information while on the go, and 70% are willing to share their location for something in return [28]. A recent study found that geofencing doubled engagements with retail stores, increasing awareness and customer visits [24]. Another marketing study with a grocery store chain and Starbucks found a 60% increase in-store visits post-campaign exposure [24].

Military service members are also plugged into their smartphones, with over 60% of their digital content views on their smartphones [29]. Geofencing and Bluetooth beacon messaging are conducive to military installations because the demographic and geolocations of service members are relatively well defined, making targeted promotion easier. Therefore, this population and setting is a prime unit for testing the feasibility of this type of technology.

Recently, a text-message-based intervention in a non-military environment aimed to reduce sodium intake by sending just-in-time adaptive messages to participants when they entered a grocery store, restaurant, or home to promote behavior change related to high sodium foods. This study indicated that those receiving these just-in-time messages reduced sodium intake by −1537 mg compared to –233 mg [30]. Despite this potential, we found no published research combining geofencing and Bluetooth beacons as a multi-layered environmental change intervention approach to improve dietary behaviors in a military context.

Therefore, the purpose of our feasibility project was to understand the potential of a multi-layered complementary smartphone messaging system (“EatWellNow”) where geofencing smartphone banner advertisements and Bluetooth beacon–triggered smartphone push notifications are sent to military service members outside (banner advertisements) and inside (push notifications) the retail food site (Army dining facility, now officially referred to as “Warrior Restaurants”). The system has an associated mobile phone application and website for additional content delivery. Our study included a secondary review of a previously collected community survey, examining the process of designing and developing EatWellNow, and understanding the feasibility of implementing this system to improve dietary behaviors at a military installation.

## 2. Materials and Methods

This formative feasibility study utilized a mixed-methods approach, including the following: (1) a secondary aggregated data review of a previously collected community assessment survey of the Fort Bragg community (military service members, family members, veterans, etc.) to understand the interest in a geofencing and beacon push notification approach, (2) development of EatWellNow, (3) examining the feasibility of this approach at a military installation, and (4) field observations of the technology to determine device functionality.

### 2.1. Secondary Data Review of a Military Community Health Assessment Survey

In collaboration with the Fort Bragg Department of Public Health, we utilized the aggregate results of the Fort Bragg Community Survey, which was part of the installation’s Community Health Assessment, collected in April–May 2021. The survey was voluntary and took 10–15 min to complete. The survey was open to anyone in the Fort Bragg community, including service members and family members of a current service member, military retirees and family members of military retirees, civilian employees, contractors, or others. Survey promotional materials were distributed through members of the Community Survey Working Group (heads of offices; individuals who informed of the creation of the survey), who were asked to share with their networks. The survey was publicized through the Womack Army Medical Center and Garrison Public Affairs offices. They posted weekly on Facebook and through their social media platforms (Facebook/Facebook groups are a significant way for Fort Bragg to communicate with social groups on post and with families). Cumberland County (where Fort Bragg is largely located) paid for Facebook promotions of the survey through the whole month of April to distribute to the county. Fort Bragg Public Health Leaders had flyers printed (with a QR code to the survey) and passed them out at their vaccine drive-throughs on post (April 2021). Individuals could take the survey while they were sitting in cars for their 15 min observation period.

The survey included questions about sociodemographic information, including biological sex, civilian or military, age (age range in years), race/ethnicity, marital status, the highest level of education completed, and military rank or sponsor’s military rank.

Respondents were asked if they would be interested in receiving push notifications about healthy foods on post and if they would be more likely to buy healthy food if they received these push notifications. They were also asked if they would be more likely to buy healthy food if they received advertisements regarding healthy foods on post within websites and apps they already use. Items related to interest in receiving notifications/messages were rated using a 4-point Likert-type scale that ranged from 1 = strongly disagree to 4 = strongly agree. The survey was open on 1 April 2021 and closed on 15 May 2021. Aggregated survey data were summarized through statistical analysis. This work is exempt through DHHS 46.101 (b) relating to unidentifiable survey or interview data for research and demonstration projects that are conducted by or subject to the approval of department or agency heads (Fort Bragg Department of Public Health), which are designed to study, evaluate, or otherwise examine public benefit or service programs (reference: DHHS, Code of Federal Regulations TITLE 45, 2009, available at https://www.hhs.gov/ohrp/regulations-and-policy/regulations/45-cfr-46/#46.101 (accessed on 13 January 2022).

### 2.2. Partnership Development

For this interactive food environment project to be successful, we initiated a collaborative partnership between the project team and relevant expert stakeholders on post, including public health officials, health promotion/wellness officials, and food service management. We partnered with a digital marketing firm (Propellant Media [31]) that focused on geotargeting advertising for digital marketing expertise and implementation of cellular geofencing. To develop and implement the Bluetooth beacon-based geofencing system, the project team partnered with an independent software programmer with expertise in developing a mobile phone application for beacon geofencing. Each of these partners had a crucial role in helping to guide the development of this interactive food environment approach.

### 2.3. Development of EatWellNow Geotargeted Messaging System

Our goal was to create a complementary multi-layer system called EatWellNow. Cellular data signal geofencing banner advertisements were the first layer of interaction with service members outside the retail food site. Bluetooth beacon messaging was the second layer of interaction with service members, taking place within the retail food environment.

#### 2.3.1. Cellular Data Signal Geofencing Banner Advertisements

The first layer of interaction is with the virtual geofence set around the location of the retail food site. When a person crosses into a cellular data signal-based virtual fence, they begin passively receiving advertisements for the retail food site on smartphone apps and websites when on their mobile phone within the food site and when they leave the virtual fence, for up to 30 days. Stakeholders from Nutrition Care (dining services), public health, the project team, and the digital marketing company met to develop smartphone geofencing advertisements. These planning meetings focused on developing messages and images for banner advertisements, the radius of the geofencing target areas, and the campaign’s duration. The goal was to create messages and use imagery relevant to military service members using the retail food site. The study team also created the geofence radius around the retail food site to efficiently target potential customers. The geofence radius was determined in collaboration with the marketing company and based on the goal of reaching soldiers using the specific commercial area where the retail food site was located and those traveling in proximity to the venue via surrounding transportation networks. This is a passive advertising approach; as long as users of the smart phone devices opt in to the location services on their devices, they receive the banner advertisements. Due to the strategic placement of the geofence, those receiving these messages are affiliated in some way with the military. No identifiable information was collected from users. We implemented the cellular data geofencing campaign for 30 days, a time period decided on in collaboration between the study team and marketing agency as a reasonable period to see consumer interaction with the messages.

#### 2.3.2. Website Landing Page

We developed a dedicated website landing page to learn more about the facility in collaboration with our stakeholders. The website could be accessed when a person clicked on either the geofenced banner ads or the beacon push notifications. The website could also be accessed directly through the web address but was not made available for search engines for this feasibility study. This website aimed to provide users with information about the military installation’s healthy venues and encourage healthier eating decisions. This website had information about the retail food site (hours of operation, location, and menus), as well as subpages on how to eat for “Athletic Performance” and “Heart Health” and create healthy meals and salads (“Healthy Meals”, “Salad Bar”). We developed website content based on input from our stakeholders’ expertise in the interests of military service members and healthy food promotion items at the retail food site.

#### 2.3.3. Bluetooth Beacons

After several meetings with relevant stakeholders to discuss the development of the system, our research team and stakeholders developed an interactive food environment Bluetooth signaling system. The goal of the system was to nudge users towards healthier decisions at each decision point with simple messaging and graphics.

The project team and food service management personnel collaborated with a software programmer to develop an interactive Bluetooth messaging system for sending healthy nudges within the retail food site. This interactive experience was based on a smartphone app we developed, called “Eat Well Now”, which served as the interaction hub between the user and the beacons. The beacons were a relay device to the user’s phone. First, a user downloads the EatWellNow app, which our team developed as a dedicated mobile phone application for the beacon experience. After downloading the app and coming within a certain distance (for this study, 10 feet) of the beacons placed in different locations within the retail food site (e.g., entrance, checkout, soda machine), the user can receive push notifications on their phone from the programmed Bluetooth beacons. There is a recommended minimum of 3 feet of spacing between the beacon and the customer for a Bluetooth beacon approach, but beacons do not interfere with each other, as they are programmed to “wait their turn”. The entire system is shown in Figure 1.

The project team met with the Army Department of Public Health, Army Health and Wellness, and food service staff to design the messages and determine the placement of beacons in the retail food site. The criteria for developing the messages were (1) to tailor each message to the location within the retail food site where behaviors were to be changed, (2) simple messaging that was quickly interpretable while shopping and that was goal oriented, and (3) to make messages relevant to a military audience. Each beacon had a specific message notification to send to the user’s phone (e.g., “Plate healthy enough? Click Here”). When users click on the message notification, they are routed to an app screen specific to that beacon location (see Figure 2).

### 2.4. Formative Feasibility Study Assessment Methods

We measured the success of the proposed project through outcomes of the process and formative evaluation measurement. We tracked the timeline to implement the technology and compared it to the proposed timeline using records of project progress. The geofencing analytics identified clicks into the advertisement, click-through rates (the percentage of people visiting a web page who access a hypertext link to a particular advertisement), the timing of clicks, the types of devices used, the topic category of websites where users were seeing and clicking the advertisements (e.g., Hobbies and Special Interests, Arts and Entertainment, etc.). The website analytics provided information on clicks into the website, time spent on the website, and features of the website where users showed the most interest through clicks and time spent. We tracked the resources needed and the amount of scheduled time allowed for the program’s implementation, including resources needed in terms of equipment, personnel, and time, using project records (logs, budget expenses, reports, etc.). We informally documented (without use of a structured, validated instrument) institutional willingness, motivation, and capacity to carry through project-related tasks, including documenting challenges and resources for fulfilling research commitments using unstructured project records and funding agency reporting materials. For management assessment, we assessed challenges and strengths of research team capacities through project records and funding agency reporting materials. For scientific assessment, we assessed if the procedures that were used protected respondent privacy and prevented potential threats to validity through monitoring of data collection and data storage. Descriptive statistics, including frequencies, means, and standard deviations, were generated for all relevant data, particularly the geofencing and website analytics data.

For this formative feasibility study, we examined the feasibility of the beacon system within the project team and not the end-users due to a restriction on implementation stated in the funding mechanism. Feasibility was based on the ability to develop and implement the system within the existing military food service infrastructure, demand for this type of approach from potential consumers, meeting market needs of food service and public health stakeholders interested in implementing the approach long term, the cost of the approach, and the ability to implement in a timely manner.

## 3. Results

### 3.1. Summary of Community Needs Assessment Survey Findings

There was a total of 3281 respondents for the community needs assessment survey. A majority (64.5%) of respondents were interested in receiving push notifications to their phones about healthy options on post. The majority (68.3%) also agreed that receiving these notifications would help them to eat healthier. Most respondents (74.6%) agreed that if they received advertisements within websites and apps they already use on their phones about healthy options on post, they would be more likely to buy healthy food. Most were interested in receiving push notifications about healthy options on post at least once per day (51%). A summary of participant demographics and survey findings can be found in Table 1.

### 3.2. Implementation

We were able to successfully deploy and make the beacons and associated app operational within the retail food site. We also created a connected data server that allowed the project team to understand user interaction with the beacons. The server was cloud-based, created through the Google Firebase SDK mobile application development platform used to develop the EatWellNow mobile app. In the future, this will allow the team to examine user engagement based on different messaging approaches.

### 3.3. Timeline Assessment

The team developed messages, geofencing advertisements, the landing page website, and the Bluetooth beacon programs within a nine-month period, which was within the study period, despite disruptions related to COVID-19.

### 3.4. Usage

For the cell signal geofencing notifications, we had 587 clicks on the advertisements in one month. The overall clickthrough rate was 0.11%, with the highest being 0.84% on day one. Clicks into the advertisement remained steady during the 30 days, with a mean of 19 clicks per day (SD = 7.73). Most clicks (79.3%) were from cell phones, followed by tablets (16.7%) and desktops/laptops (3.9%). Most of the clicks into the geofencing advertisement occurred on websites in the “Hobbies and Special Interest” (35.9%) contextual category, followed by “Arts and Entertainment” (17.9%) and “Computer and Video Games” (6.1%).

The website analytics demonstrated 703 site visits (mean of 29.5 per day, SD = 12.9) and 578 unique visitors during the one-month test period. Most (70%) of the sessions were from mobile phones. The most visited subpages were the home page, athletic performance tab, and healthy meals table. The mean time spent on the site was 12 min and 5 s.

We successfully tested the beacon system among the project team. Specifically, the project team set the beacons within the retail food environment and went through the beacon system to ensure that each beacon sent the appropriate message to mobile smartphones. We also tested clicking through the notifications to the infographics and the website to ensure they worked within the retail food site. All aspects of the beacon messaging system worked successfully within the retail food environment.

### 3.5. Resources Assessment

The resources shared between the various stakeholders meant that all the necessary resources to complete the project were available. The grant mechanism provided funds to purchase the beacons ($100 for a pack of four beacons), build the app ($3000), and purchase the cellphone-based geofencing ($250 for setup, $300 for creative asset development, and $4400 for 550,000 total impressions). The team met weekly over the project period to plan and discuss progress and had four dedicated hour-long meetings for message development and planning around beacon placement. Development of the EatWellNow app and beacon system took 200 h of dedicated time from the programmer. Stakeholders discussed strategies for having adequate resources for this approach’s long-term sustainability, including integration into dining services and public health efforts.

### 3.6. Institutional Willingness

We found high institutional willingness across relevant stakeholders across the military installation. This included Army Garrison leadership, who gave approval for the project and its implementation; food service leadership, who met with us regularly and allowed testing of the program at retail sites; and public health and health and wellness partners, who regularly attended meetings and were engaged in development and implementation. One challenge was determining action steps for obtaining approval for integrating new communication devices in the installation, but this was resolved through communication with relevant project stakeholders.

### 3.7. Management Assessment

Our team had all the necessary expertise to complete the study. The diversity of having military, academic, food service, nutrition, computer science, and communication expertise was a strength of the project. We used a graduate student computer programmer, given our budget constraints. Future development may utilize a professional programmer to expedite the process.

### 3.8. Scientific Assessment

We ensured that all processes protected the privacy of end-users. We developed and incorporated a legal privacy statement for the beacon system mobile phone application. We did not conduct direct human subject research, but we stored the data on clicks into advertisements, websites, and push notifications on a secure server.

## 4. Discussion

This study found that an interactive food environment experience using a multi-layered geofencing approach is feasible on a military installation. This feasibility is based on the ability to develop and implement the EatWellNow approach within the existing military food service infrastructure, the demand for this type of approach from potential consumers, the market of food service and public health stakeholders interested in implementing the approach long term, the relatively low cost of the approach, and the ability to implement in a timely manner. The EatWellNow approach was successful due to the participatory approach of including strategic partners across multiple relevant sectors on and off post, including in the areas of public health, nutrition, dining services, computer science, and marketing. This aligns with the literature suggesting the importance of collaboration in public health intervention work [32]. Future work should more closely examine how these cross-sector partnerships can help facilitate the modernization of military approaches to positively impact daily life activities.

Based on the community survey findings, individuals were interested in receiving app push notifications on their phones about healthy options on post. They also agreed that receiving these notifications would help them eat healthier. These results align with previous research showing that a geofencing campaign may positively influence dietary habits among adults [29] and with the larger literature around the effectiveness of mobile phone apps to promote healthier dietary and physical activity behaviors [33]. Our findings also align with a recent meta-analysis that found that nudging approaches for food choices can be particularly effective [22]. More specifically, this analysis found that decision structure (e.g., changing defaults, adjusting physical or financial efforts, social consequences, and micro-incentives) is a more impactful approach than other approaches such as decision assistance (e.g., providing reminders) or decision information (e.g., providing social norms, providing information) [22]. Future testing of EatWellNow will integrate more decision structure approaches and test the effectiveness of the different nudging approaches in the military setting. Industry reports suggest that 93% of active-duty military either visited a website, researched a product, or bought the product after they had seen an on-post advertisement for that product [34]. This suggests a desirable approach and that further testing is needed.

Customer interaction with the cell signal geofencing advertisement suggests it may be a viable approach to promote healthy eating on a military installation. Clicks represent the customer’s willingness to engage with our healthy eating advertisement. The clickthrough rate (the percentage of people on a website displaying the healthy eating advertisement who clicked the healthy eating advertisement link for more information) was within the industry average [35]. We used a singular general health promotion message of “Eat well, perform well” for the cellular geofencing over one month, as the primary goal was to test implementation. The presentation of this message was used in multiple message formats that varied in size, shape, and accompanying images. Using a singular message rather than a variety of messages may explain the relatively high clickthrough rate on day one and the lower average the rest of the month. Those who were initially attracted to the message and clicked through likely didn’t feel the need to click again with the same message and website landing page. This is reflected in the fact that most of the visitors to the website landing page were unique visitors rather than repeat visitors. The retail food site where this was tested currently serves around 20,000–30,000 customers per week. It is unclear whether the website visitors were existing or potential customers, or how much of the available consumer market would use this venue given its characteristics and location. Examining this approach at a variety of venues in different geographic areas, with different store characteristics and varying market potential, may elucidate the broader impact of this approach. Future work should examine the changes in the number of meals served and new customers using this and other retail food sites due to the geofencing messaging. Including messages around sales promotions of healthy items, as well as having a less static landing page, may be even more impactful and create sustained use. Sincemost clicks occurred on the “Hobbies and Special Interests”, “Arts and Entertainment”, and ‘Computer and Video Games” contextual categories, these may be strategic advertising contexts to reach military service members with this type of intervention. The higher interest in the “Athletic Performance” and “Healthy Meals” section suggests that these topics may be of particular interest to this audience. More work should be done to better understand use of the website, including how the use may evolve when the food environment is represented more comprehensively and how the website could be optimized to increase engagement and sustained use in conjunction with the geofencing and geotargeted messaging.

The strengths of this study include the strategic development of a multi-sector partnership to develop and assess the feasibility of this approach, the use of a structured feasibility assessment approach, community engagement through the use of a community survey with a relatively large sample, and the use of digital marketing analytics to understand interaction with the geofencing and website landing page. The limitations include the convenience sample for the community survey and the lack of direct testing of the beacon system with end-users due to funding mechanism restrictions regarding project implementation.

## 5. Conclusions

This project demonstrates a clear opportunity to build a scalable and impactful digital healthy eating interactive food environment experience to encourage military service members to eat healthier on a military installation. The involvement of the military dining services and public health collaborators in this feasibility study demonstrates the collective interest in and potential of using this approach on a military installation. We believe that there are many opportunities to further develop the interactive experience, including enhancing the app interface to allow for more tailored content based on the soldiers’ goals and integrating the geofencing and Bluetooth analytics to create a cohesive food environment experience. The project’s next steps are to further develop the system, with direct feedback from military service members, to optimize relatability and usability through focus groups and iterative user-centered testing, and then examine the impact of the program through an efficacy trial. Future work will examine the impact of the intervention on food purchases, dietary behaviors, and health outcomes among military service members interacting with the intervention.

## Figures and Tables

**Figure 1 nutrients-14-01458-f001:**
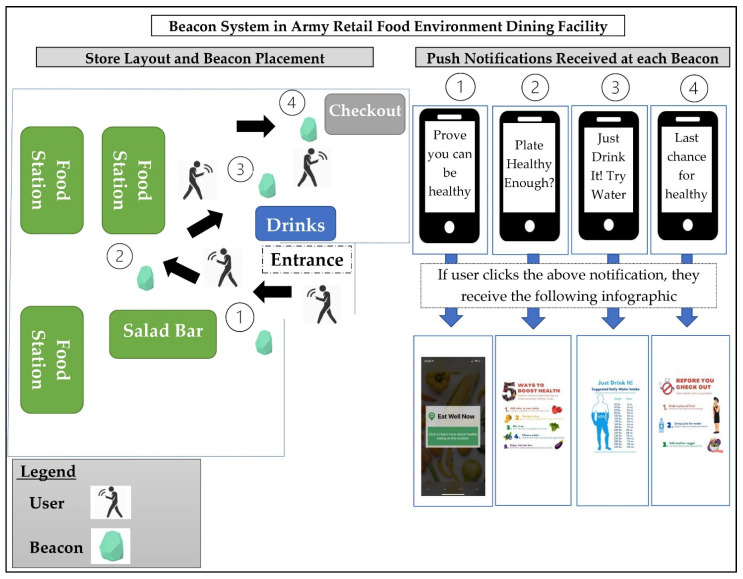
Map of the Bluetooth beacon setup within the food shopping environment and the corresponding messages and infographics one would see at each beacon location.

**Figure 2 nutrients-14-01458-f002:**
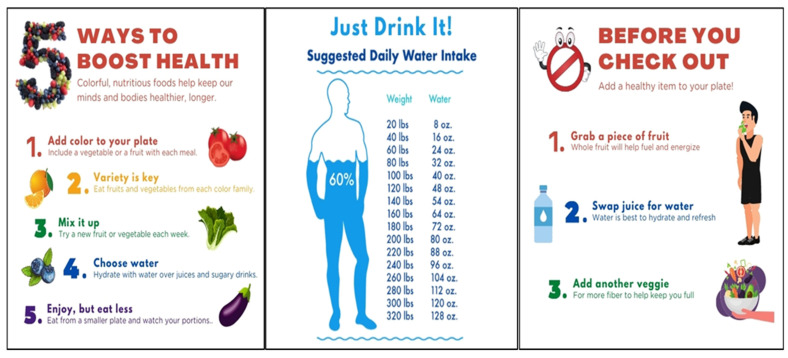
Healthy eating infographics the user would see if they clicked into the Bluetooth beacon push notifications they received while in the interactive food purchasing environment.

**Table 1 nutrients-14-01458-t001:** Summary of community needs assessment survey participant characteristics and findings regarding interest in receiving push notifications and potential impact on dietary habits.

Attribute	% (*n*)
Sex	
Male	45.3% (1187)
Female	54.7% (1434)
Age (year)	
18–25	24% (645)
26–39	53.7% (1444)
40–54	16.4% (442)
55+	5.8% (157)
Race	
White	64.6% (1747)
Non-White	35.4% (956)
Ethnicity	
Latino	47.7% (1198)
Non-Latino	52.3% (1314)
Relationship to Fort Bragg	
Active Service Member	19.9% (653)
Family of Active Service Member	37.7% (1237)
Military Retiree and Family	21.9% (720)
Civilian Employee	16.1% (527)
Contractor	2.9% (95)
Other	1.5% (49)
Military Rank	
Officer	50.3% (1247)
Non-Officer (Enlistee)	49.7% (1230)
Interested in receiving push notifications to their phones about healthy options on post ^a^	64.5% (1697/2589)
Receiving these notifications would help them to eat healthier ^b^	68.3% (1789/2584)
Agreed that if they received advertisements within websites and apps they already use on their phones about healthy options on post, they would be more likely to buy healthy food ^c^	74.6% (1942/2671)

^a^ Original Question (4-point Likert response of strongly disagree to strongly agree): “I would be interested in receiving push notifications/messages to my phone about healthy options on post”. ^b^ Original Question (4-point Likert response of strongly disagree to strongly agree): “If I received push notifications/messages to my phone about healthy options on post, I would be more likely to buy healthy food”. ^c^ Original Question (4-point Likert response of strongly disagree to strongly agree): “If I received advertisements on websites and apps I already use about healthy options on post, I would be more likely to buy healthy food”.

## Data Availability

Data are available upon request from the corresponding author.

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
