# Peer review of "EatWellNow: Formative Development of a Place-Based Behavioral “Nudge” Technology Intervention to Promote Healthier Food Purchases among Army Soldiers"

_nutrients, 2022, doi:10.3390/nu14071458_

Round 1

Reviewer 1 Report

Thank you for allowing me the chance to read your interesting article which tests the use of geofencing and Bluetooth beacons in nudging military personnel to make healthier food choices. The EatWellNow format has the potential to be used in other workplaces and I will be interested to read the results of future work.

I have a few suggestions, which I hope will help improve your paper.

  1. P.2, line 85: please add ‘in’ between ‘something’ and ‘return’.
  2. P.4, lines 158 and 174 (also p.5, line 204) Please can you differentiate the first sentences as a sub-heading or put in bold or italics.
  3. P.6, lines 252-254. This information about Hobbies, arts etc seem to come from nowhere. There is no previous mention of these types of ads, only information about food choices and sales. Please can you provide more information about all the ads that a user will see to give more context to the results.
  4. P.8. The informed consent statement, Data availability statement and Acknowledgments sections need to be populated accordingly.

Author Response

Thank you to the reviewers and Editorial staff for your time and effort and reviewing this manuscript. Your wonderful feedback is greatly appreciated! We are thankful for your feedback and have tried to make all adjustments to the paper as suggested. We are amenable to additional changes if necessary. Thank you for your consideration of our manuscript!

Reviewer

Reviewer Comment

Response

Reviewer 1

P.2, line 85: please add ‘in’ between ‘something’ and ‘return’.

We added “in”.

Reviewer 1

P.4, lines 158 and 174 (also p.5, line 204) Please can you differentiate the first sentences as a sub-heading or put in bold or italics.

We have changed to align with journal formatting recommendations for this level of headings.

Reviewer 1

P.6, lines 252-254. This information about Hobbies, arts etc seem to come from nowhere. There is no previous mention of these types of ads, only information about food choices and sales. Please can you provide more information about all the ads that a user will see to give more context to the results.

We added the following to the Methods:

Lines 258-262: “The geofencing analytics identified clicks into the advertisement, click through rates (the percentage of people visiting a web page who access a hypertext link to a particular advertisement), timing of clicks, the types of devices used, the topic category of websites where users were seeing and clicking the advertisements (e.g., Hobbies and Special Interests, Arts & Entertainment, etc.).”

Reviewer 1

P.8. The informed consent statement, Data availability statement and Acknowledgments sections need to be populated accordingly.

We have populated accordingly.

Reviewer 2 Report

The manuscript summarizes a feasibility study leveraging novel technologies intended to nudge healthier eating among soldiers. Overall, the concepts are important, but the authors leave out several key details, which are critical for a making compelling case. The biggest concerns with this study are the lack of details regarding the methodology and intervention, failure to explain in sufficient the results, lack of any data presented to support conclusions for resource assessment, management assessment, and scientific assessment. I encourage the authors to consider referencing primary peer-reviewed publications when possible rather than relying on news stories, blogs, one-page fact sheets, and Google books.

Specific comments follow.

Abstract:

Line 24 – A brief description of what EatWellNow is would be helpful

Introduction:

Lines 35-36 – How is obesity defined? Based on BMI (≥ 30 kg/m2) or body composition, presumably based on Army standards?

Line 44– Clarify the proportion that “most” military service members represents.

Line 51 – A general textbook citation (reference 11) is does not appear to support your statement related to the military.

Line 53 – Have there been any formal or peer-reviewed publications evaluating food offerings on military bases rather than the current citation.

Line 56 – Clarify what is “significant,” how was this defined?

Line 69 – There appears to be a word missing, sentence is unclear.

Lines 70-73 – Consider breaking up into multiple sentences to improve readability.

Methods:

Lack of details regarding formative evaluation described in the abstract.

How as the Fort Bragg Community Survey disseminated?

What digital marketing firm did you partner with?

Who developed the Bluetooth beacon-based geofencing system?

How do you ensure your messaging system only targets service members? Did service members provide their consent to be targeted? Where did you recruit the service members from? Any demographic information on those service members that participated?

Was there any pilot testing done to ensure messages and imagery were relevant to military service members prior to using then in EatWellNow

What is the rationale for selecting a 30-day geofencing campaign? Is this the industry standard?

How did service members learn about the app?

No mention of statistical analysis for analytics data.

Results:

Use a table to summarize results from the Fort Bragg community health assessment.

What was the response rate for the Fort Bragg Community Survey? What was the demographic breakdown? If these results have been published elsewhere, please include a citation.

Do you have any data on how much time people spent interacting with the app or the website?

Line 242 – How was effectiveness defined?

Line 247 – Encourage the authors to use standard statistical documentation when reporting means and standard deviations.

Discussion:

There is an overall lack of commentary on the interpretation of your results, how your results agree or disagree with previous publications, and the implications and practical applications of your work.

For a reader not familiar with analytics, please provide an interpretation of what the number of clicks means and how this is relevant.

What do you think contributed to a clickthrough rate of 0.84% on one day, when the average was 0.11%

Lines 284-285 – What is this statement based of off?

Lines 294-295 – Specifically, which groups were more interested in technology.

Figures:

Additional details are needed, as is they are not stand alone items.

Author Response

Thank you to the reviewers and Editorial staff for your time and effort and reviewing this manuscript. Your wonderful feedback is greatly appreciated! We are thankful for your feedback and have tried to make all adjustments to the paper as suggested. We are amenable to additional changes if necessary. Thank you for your consideration of our manuscript!

Reviewer

Reviewer Comment

Response

Reviewer 2

Lack of details regarding the methodology and intervention, Failure to explain in sufficient the results, lack of any data presented to support conclusions for resource assessment, management assessment, and scientific assessment

Thank you for this feedback. We have tried to address each of these summary points within the specific responses listed below.

Reviewer 2

I encourage the authors to consider referencing primary peer-reviewed publications when possible rather than relying on news stories, blogs, one-page fact sheets, and Google books.

Thank you for this suggestion. We have tried to replace non-peer reviewed blogs, fact sheets, and books with peer reviewed publications where possible. Due to this being an innovative field with little published research, there is some reliance on industry data. Please let us know if there are specific citations that should be removed prior to publication.

Reviewer 2

Abstract:

Line 24 – A brief description of what EatWellNow is would be helpful

We added the following:

Lines 25-27: “a multi-layered interactive food environment approach using contextual messaging to improve food purchasing decisions within the military food environment context.”

Reviewer 2

Introduction:

Lines 35-36 – How is obesity defined? Based on BMI (≥ 30 kg/m2) or body composition, presumably based on Army standards?

We added the following:

Lines 38-41: “A 2020 report suggests that one in five military service members are classified as obese (BMI of 30.0 and above) and have difficulty meeting the Army Body Composition Program (ABCP) standard (BMI of 27.5 or lower, or body fat percentage of 26 percent for men and 36 percent for women service members).”[1]

Reviewer 2

Line 44– Clarify the proportion that “most” military service members represents.

We removed this since this information was already presented in Line 1, and replaced it with this:

Lines 48-49: “Research suggests that Soldiers have low adherence to the Dietary Guidelines for Americans. [5]”

Reviewer 2

Line 51 – A general textbook citation (reference 11) is does not appear to support your statement related to the military.

We have removed this statement and citation as it was unnecessary.

Reviewer 2

Line 53 – Have there been any formal or peer-reviewed publications evaluating food offerings on military bases rather than the current citation.

We have added the following:

The Healthy Base Initiative | Bipartisan Policy Center Available online: https://bipartisanpolicy.org/report/healthy-base-initiative/ (accessed on 10 November 2021).

Troncoso, M.R., Jayne, J.M., Robinson, D.J. and Deuster, P.A., 2021. Targeting nutritional fitness by creating a culture of health in the military. Military Medicine, 186(3-4), pp.83-86.

Shams-White, M.M., Cuccia, A., Ona, F., Bullock, S., Chui, K., McKeown, N. and Must, A., 2019. Lessons learned from the creating active communities and healthy environments toolkit pilot: a qualitative study. Environmental health insights, 13, p.1178630219862231.

Reviewer 2

Line 56 – Clarify what is “significant,” how was this defined?

We changed to:

Lines 60-63: “While improving availability is essential, the service members must be aware of the availability and feel accommodated during their food purchase (i.e., the environment is set up to make shopping easy and enjoyable).[20]”

Reviewer 2

Line 69 – There appears to be a word missing, sentence is unclear.

We changed to the following:

Lines 74-75: “These virtual perimeters can be established in multiple ways, including using cellular data signals. “

Reviewer 2

Lines 70-73 – Consider breaking up into multiple sentences to improve readability.

We simplified the sentence to the following:

Lines 75-78: “A person crosses into a virtual geofence with their mobile device (i.e., smartphone). If the location service on their device is on (in an app or website that they are on), the person's location can be detected to receive the geofence messaging.”

Reviewer 2

Methods:

Lack of details regarding formative evaluation described in the abstract.

We added the following to the Methods:

Lines 255-256: “We tracked the timeline to implement the technology and compared it to the proposed timeline using records of project progress.”

Lines 258-262: “The geofencing analytics identified clicks into the advertisement, click through rates (the percentage of people visiting a web page who access a hypertext link to a particular advertisement), timing of clicks, the types of devices used, the topic category of websites where users were seeing and clicking the advertisements (e.g., Hobbies and Special Interests, Arts & Entertainment, etc.).”

Lines 262-264: “The website analytics provided information on clicks into the website, time spent on the website, and features of the website where users showed the most interest through clicks and time spent.”

We also clarified that the resources needed, institutional willingness, management assessment, and scientific assessment were tracked using project records (logs, budget expenses, reports).

Reviewer 2

How as the Fort Bragg Community Survey disseminated?

We added the following:

Lines 131-142: “Survey promotional materials were distributed through members of the Community Survey Working Group (heads of offices; individuals who informed the creation of the survey); they were asked to share with their networks. The survey was publicized through the Womack Army Medical Center and Garrison Public Affairs offices – they posted weekly on Facebook and through their social media platforms. (Face-book/Facebook groups is a huge way that Ft Bragg communicates with social groups on base and with families). Cumberland County (where Fort Bragg is largely located) paid for Facebook promotions of the survey through the whole month of April to distribute to the county. Ft Bragg Public Health Leaders had flyers printed (with a QR code to the survey) and passed them out at their vaccine drive-throughs on base (April 2021). Individuals could take the survey while they were sitting in cars for their 15-minute observation period.”

Reviewer 2

What digital marketing firm did you partner with?

We added the following:

“Propellant Media [31]”

Reviewer 2

Who developed the Bluetooth beacon-based geofencing system?

We changed to the following:

Liens 167-168: “…the project team partnered with an independent software programmer with expertise in developing a mobile phone application for beacon geofencing.”

Further down, we also state this:

Lines 210-216: “After several meetings with relevant stakeholders to inform the development of the system, our research team and stakeholders developed an interactive food environment Bluetooth signaling system. The goal was to nudge users towards healthier decisions at each decision point with simple messaging and graphics.

                The project team and foodservice management personnel collaborated with a software programmer to develop an interactive Bluetooth messaging system for sending healthy nudges within the retail food site.”

Reviewer 2

How do you ensure your messaging system only targets service members?

We added the following:

Lines 191-194: “This is a passive advertising approach, so as long as users of the smart phone devices opt-in to the location services on their devices, they would receive the banner advertisements. Due to the strategic placement of the geofence, those receiving these messages are affiliated in some way with the military.” 

Reviewer 2

Did service members provide their consent to be targeted?

We added the following:

Lines 191-194: “This is a passive advertising approach, so as long as users of the smart phone devices opt-in to the location services on their devices, they would receive the banner advertisements.”

Reviewer 2

Where did you recruit the service members from?

Service members were not recruited, and no identifiable information was collected. As long as users of the smart phone devices opt-in to the location services on their devices, they would receive the banner advertisements:

Lines 191-194: “This is a passive advertising approach, so as long as users of the smart phone devices opt-in to the location services on their devices, they would receive the banner advertisements. No identifiable information was collected from users.”

Reviewer 2

Any demographic information on those service members that participated?

We added the following regarding the cellular geofencing:

Liens 194-195: “No identifiable information was collected from users.”

Reviewer 2

Was there any pilot testing done to ensure messages and imagery were relevant to military service members prior to using then in EatWellNow

This study did not do direct pilot testing with consumers due to restrictions on end-user engagement within the funding mechanism, but that is the next step now that we have developed the system and are on to the next phase of funding. We did have the following in the manuscript describing our process, which relied on topical experts:

Lines 181-186: “Stakeholders from Nutrition Care (dining services), public health, the project team, and the digital marketing company met to develop smartphone geofencing advertisements. These planning meetings focused on developing messages and images to be used in the banner advertisements, the radius of the geofencing target areas, and the campaign’s duration. The goal was to create messages and use imagery relevant to military service members using the food venue where testing.”

We also added this:

Lines 385-388: “The limitations include the convenience sample for the community survey, and the lack of direct testing of the beacon system with end users due to funding mechanism restrictions regarding project implementation.    

Reviewer 2

What is the rationale for selecting a 30-day geofencing campaign? Is this the industry standard?

We added the following:

Lines 195-198: “We implemented the cellular data geofencing campaign for 30 days, a time period decided on in collaboration between the study team and marketing agency as a reasonable period to see consumer interaction with the messages.”

Reviewer 2

How did service members learn about the app?

We added the following to the manuscript to clarify:

Lines 240-242: “For this formative pilot study, the beacon system was only tested for feasibility and functionality within the project team and not end-users due to a restriction on implementation stated in the funding mechanism.”

Reviewer 2

No mention of statistical analysis for analytics data.

We added the following to the manuscript:

Lines 274-276: “Descriptive statistics, including frequencies, means, and standard deviations, were generated for all relevant data, particularly the geofencing and website analytics data.”

Reviewer 2

Results:

Line 242 – How was effectiveness defined?

We removed the word “effectiveness” and changed to:

Lines 296-298: “We also created a connected data server that allows the project team to understand user interaction with the beacons. In the future, this will allow the team to examine user engagement based on different messaging approaches.”

Reviewer 2

Line 247 – Encourage the authors to use standard statistical documentation when reporting means and standard deviations.

Thank you for this suggestion. We have changed this.

Reviewer 2

Use a table to summarize results from the Fort Bragg community health assessment.

We have added a Table to summarize the characteristics of the sample and to summarize results.

Reviewer 2

What was the response rate for the Fort Bragg Community Survey? What was the demographic breakdown? If these results have been published elsewhere, please include a citation.

We have added information to the manuscript indicating that this was a convenience sample, and that the survey was broadly disseminated through advertisements, intercept recruitment, etc. We have added a Table to summarize the characteristics of the sample and to summarize results.

We also added the following:

Lines 131-142: “Survey promotional materials were distributed through members of the Community Survey Working Group (heads of offices; individuals who informed the creation of the survey); they were asked to share with their networks. The survey was publicized through the Womack Army Medical Center and Garrison Public Affairs offices – they posted weekly on Facebook and through their social media platforms. (Face-book/Facebook groups is a huge way that Ft Bragg communicates with social groups on base and with families). Cumberland County (where Fort Bragg is largely located) paid for Facebook promotions of the survey through the whole month of April to distribute to the county. Ft Bragg Public Health Leaders had flyers printed (with a QR code to the survey) and passed them out at their vaccine drive-throughs on base (April 2021). Individuals could take the survey while they were sitting in cars for their 15-minute observation period.”

Reviewer 2

Do you have any data on how much time people spent interacting with the app or the website?

We have added the following:

Lines 317-318: “The mean time spent on the site was 12 minutes and 5 seconds.”

Reviewer 2

Discussion:

There is an overall lack of commentary on the interpretation of your results, how your results agree or disagree with previous publications, and the implications and practical applications of your work.

Thank you for this feedback. While this is a relatively novel project with not much literature to compare it to, we have tried to bolster the Discussion section with the following:

Lien 349-351: “This aligns with literature suggesting the importance of collaboration in public health intervention work [27].”

Lines 357-360: “These results align with previous research showing that a geofencing campaign may positively influence dietary habits among adults, [30] and with the larger literature around the effectiveness of mobile phone apps to promote healthier dietary and physical activity behaviors.[33]

Lines 372-375: “Using a singular message rather than a variety of messages may explain the relatively high clickthrough rate on day 1 and the lower average the rest of the month. Those who were initially attracted to the message and clicked through likely didn’t feel the need to click again with the same message and landing page. Including messages around sales promotions of healthy items, as well as having a less static landing page. may be even more impactful.”

Lines 366-371: “Customers' interaction with the cell signal geofencing advertisement suggests that it may be a viable approach to promote healthy eating on the military base. Clicks represent the customer’s willingness to engage with our healthy eating advertisement. The clickthrough rate (the percentage of people on a website displaying the healthy eating advertisement who clicked the healthy eating advertisement link for more information) was within the industry average.”

Lines 381-388: “The strengths of this study include the strategic development of a multi-sector partnership to develop and assess the feasibility of this approach, use of a structured feasibility assessment approach, community engagement through the use of a community survey with a relatively large sample, and the use of digital marketing analytics to understand interaction with the geofencing and website landing page. The limitations include the convenience sample for the community survey, and the lack of direct testing of the beacon system with end users due to funding mechanism restrictions regarding project implementation.    

Reviewer 2

For a reader not familiar with analytics, please provide an interpretation of what the number of clicks means and how this is relevant.

We added the following:

Lines 366-371: “Customers' interaction with the cell signal geofencing advertisement suggests that it may be a viable approach to promote healthy eating on the military base. Clicks represent the customer’s willingness to engage with our healthy eating advertisement. The clickthrough rate (the percentage of people on a website displaying the healthy eating advertisement who clicked the healthy eating advertisement link for more information) was within the industry average.”

Reviewer 2

What do you think contributed to a clickthrough rate of 0.84% on one day, when the average was 0.11%

We added the following:

Lines 371-377: “We used a singular general health promotion message of "“Eat well, perform well"”. Using a singular message rather than a variety of messages may explain the relatively high clickthrough rate on day 1 and the lower average the rest of the month. Those who were initially attracted to the message and clicked through likely didn’t feel the need to click again with the same message and landing page. Including messages around sales promotions of healthy items, as well as having a less static landing page. may be even more impactful.

Reviewer 2

Lines 284-285 – What is this statement based of off?

We added the following:

Lines 344-346: “This study found that an interactive food environment experience using a multi-layered geofencing approach is feasible within the military base setting based on the ability to develop and implement this approach in a feasible manner.”

Reviewer 2

Lines 294-295 – Specifically, which groups were more interested in technology.

We removed this from the manuscript. We collected demographic data as part of the needs assessment community survey, but are only able to report summary findings.

Reviewer 2

Figures:

Additional details are needed, as is they are not stand alone items.

We added more details to the figure titles:

Lines 237-239: “Figure 2. Healthy eating infographics the user would see if they clicked into the Bluetooth beacon push notifications they received while in the interactive food purchasing environment.”

Round 2

Reviewer 2 Report

The authors have made considerable updates to the original manuscript. However, the manuscript is still lacking considerable detail. The manuscript lacks focus as the authors refer to the study as both a feasibility project (line 109) and formative pilot study (line 240). The majority of this paper is focused on methods with limited results. For example, Section 3.7 states, "We ensured that all processes protected the privacy of end-users." Based on the methods section, it is unclear to the reader how this was accomplished and subsequently the results that support this statement. I suggest the authors consider submitting as a short communications. Specific comments follow: 

Introduction:

Consider recent publication on nudging and choice architecture (https://www.ncbi.nlm.nih.gov/pmc/articles/PMC8740589/) 

Lines 40-41 - Consider deleting statement about Army Body Composition Standards. These standards vary based on sex and age. The example provided, BMI 27.5 applies to men only and body fat percentages are for men and women >40 years of age. The percentage is lower for younger individuals. 

The words base, post, and installation are used throughout. Consider using only one term to describe the setting.

Methods:

Is the food venue and food retail site the same thing? If so, is this a cafeteria, restaurant or store? 

Line 172: Please clarify what EatWellNow refers to. It appears to be the name of the program, the mobile application, and perhaps a website?

Line 200: Is the website only accessible through the geotargeted messaging system? 

Line 201-202: Please clarify who the "user" is in this sentence

Line 230: What criteria were used to create and determine messaging? Do the beacons have to be a certain distance apart so as not to interfere with one another? 

Line 240: What outcomes were used to determine feasibility? Consider including these values in the results section. 

Line 265-266: Do you have any objective data you can include here and report on regarding resources, amount of time, implementation ... etc.?

Line 267-269: Where were these tasks documented? Are you able to include them in the results section? 

Line 272: Please clarify what a project log is? Is this a standardized reporting mechanism required by the funding agency? 

Results:

Line 296: How was the connected data server created? Consider including in the methods section. 

Discussion:

Line 346: The term "feasible manner" is not clear. Does this refer to within budget and on time? 

Line 362: Please include the original source if you are able to rather than a blog. 

Line 372-374: How many messages were developed for the one month intervention? This statement implies that only one message (Eat well, perform well) was used the entire month, is this accurate? 

Do you have any thoughts on your website analytics given 703 site visitors and 578 unique visitors? Any idea how many people generally eat at this venue in a given month? Any plans for how you would change this in the future? 

Conclusion: 

Line 399-400: What is your plan for assessing relatability, usability and impact? 

Table 1: For the three statements included at the end of the table, it would helpful to see the questions as they were written on the survey. 

Author Response

Thank you for the time and effort you spent reviewing this manuscript again! We are very appreciative of your feedback. We believe that addressing your feedback has improved the overall quality of the paper.

Reviewer Comment

Response

The manuscript lacks focus as the authors refer to the study as both a feasibility project (line 109) and formative pilot study (line 240).

We have changed the wording to more accurately describe the paper as a formative feasibility study.

I suggest the authors consider submitting as a short communications

We are willing to submit as a short communication if necessary.

Introduction:

Consider recent publication on nudging and choice architecture (https://www.ncbi.nlm.nih.gov/pmc/articles/PMC8740589/)

We added the following to the Introduction:

Page 2, Lines 70-72: “A recent meta-analysis of choice architecture nudging interventions found that choice architecture interventions overall promote behaviors change, particularly when influencing food choices. [23]

We added the following to the Discussion:

Page 11, Lines 384-392: “Our findings also align with a recent meta-analysis which found that nudging approaches for food choices can be particularly effective [23]). More specifically they found that decision structure (e.g., changing defaults, adjust physical or financial effort, social consequences or micro-incentives) as more impactful approach than other approaches such as decision assistance (e.g., providing reminders) or decision information (e.g., providing social norms, providing information) [23]. Future testing of EatWellNow will integrate more decision structure approaches and test the effectiveness of the different nudging approaches in the military setting.”

Lines 40-41 - Consider deleting statement about Army Body Composition Standards. These standards vary based on sex and age. The example provided, BMI 27.5 applies to men only and body fat percentages are for men and women >40 years of age. The percentage is lower for younger individuals.

We have removed this statement to avoid confusion.

The words base, post, and installation are used throughout. Consider using only one term to describe the setting.

We have now tried to make this as consistent as possible by using “post” and “military installation”, but the military uses these terms interchangeably and as formal names (e.g. Healthy Base Initiative), so in some cases changes couldn’t be made.

Is the food venue and food retail site the same thing? If so, is this a cafeteria, restaurant or store?

Yes, these were the same thing. We have changed to make the language more consistent. To clarify, we have added the location:

Page 3, Lines 111-113: “…retail food site (Army dining facility, now officially referred to as “Warrior Restaurants”).

Line 172: Please clarify what EatWellNow refers to. It appears to be the name of the program, the mobile application, and perhaps a website?

It refers to the system and the associated mobile phone app. We have tried to clarify this in the manuscript:

Page 3, Lines 107-116: “Therefore, the purpose of our feasibility project was to understand the potential of a multi-layered complementary smartphone messaging system (“EatWellNow”) where geofencing smartphone banner advertisements and Bluetooth beacon triggered smartphone push notifications are sent to military service members outside (banner advertisements) and inside (push notifications) the retail food site (Army dining facility, now officially referred to as “Warrior Restaurants”). The system has an associated mobile phone application and website for additional content delivery.”

Line 200: Is the website only accessible through the geotargeted messaging system?

We added the following:

Page 5, Lines 205-207: “The website could also be accessed directly through the web address but was not made available for search engines for this feasibility study.”

Line 201-202: Please clarify who the "user" is in this sentence

We changed the wording to the following:

Page 5, Lines 203-204: “The website could be accessed when a person clicked on either the geofenced banner ads or the beacon push notifications.”

Line 230: What criteria were used to create and determine messaging? Do the beacons have to be a certain distance apart so as not to interfere with one another?

We added the following:

Page 6, Lines 240-242: “The criteria for developing the messages were to 1) make each message tailored to the location within the retail food site where behaviors were to be changed, 2) simple messaging that was easily interpretable while shopping and that was goal oriented, and 3) relevant to a military audience.”

Page 5, Lines 229-231: “There is a recommended minimum of 3 feet of spacing between the beacon and the customer for a Bluetooth beacon approach, but beacons do not interfere with each oth-er, as they are programmed to “wait their turn”.

Line 240: What outcomes were used to determine feasibility? Consider including these values in the results section.

We moved this statement to after the section on ‘formative feasibility study assessment methods’, and modified to the following:

Page 7, Lines 277-283: “For this formative feasibility study, we examined the feasibility of the beacon system within the project team and not end-users due to a restriction on implementation stated in the funding mechanism. Feasibility was based on the ability to develop and implement the system within the existing military food service infrastructure, having demand for this type of approach from potential consumers, meeting market needs of food service and public health stakeholders interested in implementing the approach long term, the cost of the approach, and the ability to implement in a timely manner.”

Line 265-266: Do you have any objective data you can include here and report on regarding resources, amount of time, implementation ... etc.?

We added the following:

Page 10, Lines 333-340: “The grant mechanism provided funds to purchase the beacons ($100 for a pack of four beacons), build the app ($3,000), and to purchase the cell phone-based geofencing ($250 for setup, $300 for creative asset development, and $4,400 for 550,000 total impressions). The team met weekly over the project period to plan and discuss progress and had four dedicated hour-long meetings for message development and planning around beacon placement. Development of the EatWellNow app and beacon system took 200 hours of dedicated time from the programmer.”

Line 267-269: Where were these tasks documented? Are you able to include them in the results section?

We added the following to clarify:

Page 7, Lines 265-268: “We informally documented (without use of a structured, validated instrument) institutional willingness, motivation, and capacity to carry through project-related tasks, including documenting challenges and resources for fulfilling re-search commitments using unstructured project records and funding agency reporting materials.”

We also added a new section regarding institutional willingness to the Results section:

Page 10, Lines 344-352:

“3.6 Institutional Willingness

We found high institutional willingness across relevant stakeholders across the military installation. This included Army Garrison leadership who gave approval for the project and its implementation, food service leadership who met with us regularly and allowed testing of the program at retail sites, and public health and health and wellness partners who regularly attended meetings and were engaged in development and implementation. One challenge was determining action steps for getting approval for integrating new communication devices on the installation, but this was resolved through communication with relevant project stakeholders.

Line 272: Please clarify what a project log is? Is this a standardized reporting mechanism required by the funding agency?

We tried to clarify the wording. Our team tracked progress through recording and through funding mechanism required reporting. It was not an event by event log, but an assessment of overall progress:

Page 7, Line 271: “…through project records and funding agency reporting materials.”

Line 296: How was the connected data server created? Consider including in the methods section.

We added the following:

Page 9, Lines304-307: “The server is a cloud-based server created through the Google Firebase SDK mobile application development platform used to develop the EatWellNow mobile app.”

Line 346: The term "feasible manner" is not clear. Does this refer to within budget and on time?

We changed to the following:

Page 10 Lines 365-371: “This study found that an interactive food environment experience using a multi-layered geofencing approach is feasible on a military installation. This feasibility is based on the ability to develop and implement the EatWellNow approach within the existing military food service infrastructure, the demand for this type of approach from potential consumers, the market of food service and public health stakeholders interested in implementing the approach long term, the relatively low cost of the approach, and the ability to implement in a timely manner.”

Line 362: Please include the original source if you are able to rather than a blog.

We were able to locate the webpage for the original source and modified the references, though the details are proprietary research.

“Military Explorer | Refuel Agency Available online: https://www.refuelagency.com/militaryexplorer/ (accessed on 22 March 2022).

Line 372-374: How many messages were developed for the one month intervention? This statement implies that only one message (Eat well, perform well) was used the entire month, is this accurate?

We clarified that this is regarding the cellular geofencing component:

Page 11: Lines 401-404: “We used a singular general health promotion message of “Eat well, perform well” for the cellular geofencing over a one-month period as the main goal was to test implementation. The presentation of this message was used in multiple message formats that varied by size, shape, and accompanying images.”

Do you have any thoughts on your website analytics given 703 site visitors and 578 unique visitors? Any idea how many people generally eat at this venue in a given month? Any plans for how you would change this in the future?

We added the following:

Page 11, Lines 408-417:  “This is reflected in the fact that most of the visitors to the website landing page were unique visitors rather than repeat visitors… The higher interest in the “athletic performance’ and “healthy meals” section suggests that these topics may be of particular interest to this audience. More work should be done to better understand use of the website and how use may evolve when the food environment is represented more comprehensively, and how the website could be optimized to increase engagement and sustained use in conjunction with the geofencing and geotargeted messaging.”

We also added this to the Discussion:

Page 11, Lines 410-417: “The retail food site where this was tested currently serves around 20-30,000 customers per week. It is not clear if the website visitors were existing or potential new customers, and how much of the available consumer market would use this venue given its characteristics and location. Examining this approach at a variety of venues in different geographic areas, different store characteristics, and varying market potential may elucidate the broader impact of this approach. Future work should examine for changes in the number of meals served and new customers using this and other retail food sites due to the geofencing messaging.”

Line 399-400: What is your plan for assessing relatability, usability and impact?

We added the following:

Page 12, Lines 445-450: “The project’s next steps are to further develop the system with direct feedback from military service members to optimize relatability and usability through focus groups and iterative user-centered testing, and then examine impact of the program on through an efficacy trial.”

Table 1: For the three statements included at the end of the table, it would helpful to see the questions as they were written on the survey.

We added the following as footnotes to the table:

aOriginal Question (4 point likert response of strongly disagree to strongly agree): “I would be interested in receiving push notifications/messages to my

phone about healthy options on post.”

bOriginal Question (4 point likert response of strongly disagree to strongly agree): “If I received push notifications/messages to my phone about healthy options on post, I would be more likely to buy healthy food.”

c Original Question (4 point likert response of strongly disagree to strongly agree): “If I received advertisements on websites and apps I already use about healthy options on post, I would be more likely to buy healthy food.”